# Using Light Polarization to Identify Fiber Orientation in Carbon Fiber Components: Metrological Analysis

**DOI:** 10.3390/s24175685

**Published:** 2024-08-31

**Authors:** Luciano Chiominto, Giulio D’Emilia, Emanuela Natale

**Affiliations:** Department of Industrial and Information Engineering and Economics, University of L’Aquila, 67100 L’Aquila, Italy; giulio.demilia@univaq.it (G.D.); emanuela.natale@univaq.it (E.N.)

**Keywords:** polarized camera, carbon fiber composites, uncertainty, NDT, component inspection, fiber orientation

## Abstract

In this work, a method for measuring tow angles in carbon fiber components, based on the use of a polarized camera, is analyzed from a metrological point of view. Carbon fibers alter the direction of the reflected light’s electrical field, so that in each point of the surface of a composite piece, the angle of polarization of reflected light matches the fiber orientation. A statistical analysis of the angle of linear polarization (AoLP) in each pixel of each examined area allows to evaluate the average winding angle. An evaluation of the measurement uncertainty of the method on a cylinder obtained by a filament winding process is carried out, and the result appears adequate for the study of the distribution of angles along the surface of the piece, in order to optimize the process.

## 1. Introduction

The use of carbon fiber-reinforced polymers (CFRPs) is becoming increasingly common due to the possibility of producing robust yet lightweight components, as required in many automobile or aviation applications, as well as in marine and civil engineering. The wide design space of the final material characteristics of the product, which can be highly customized, is another noticeable feature of CFRPs [1,2].

The growing usage of CFRPs has increased the demand for solutions related to inspecting such materials. In particular, the positioning and orientation of fibers during manufacturing are aspects of great interest, because they can significantly affect the strength and stiffness of composite components [3,4], but require specific inspection methods for the identification of irregularities.

A non-destructive evaluation (NDE), which is carried out at predetermined intervals, and continuous structural health monitoring (SHM), or its advanced version as prognostics and health monitoring (PHM), which also permits forecasting and avoiding any sudden critical failure, as part of the manufacturing process and during the in-service lifetime of composites, are necessary for their effective use in a wide range of applications [5,6].

The production process of composites is complex and involves several factors. As a result, there is a significant probability of various errors occurring, which heightens the importance of safety issues throughout the product’s life. This makes NDE even more important during the production process.

It should be also considered that parts inspection is a time-consuming process, so the accurate, fast, and possibly automatable, fiber orientation assessment is a desirable target for production quality control.

Filament winding is a particular production process for composite components, which involves wrapping continuous filaments coated with resin around a rotating mandrel in specific patterns, to create an axisymmetric composite structure. Once the desired thickness is achieved, the structure is cured and the mandrel is removed, leaving a hollow composite part [7,8].

In this kind of process, the winding angle represents a particularly critical parameter, as it determines the mechanical characteristics of the component, and any irregularities in the movement of the winding head are reflected in fiber orientation defects. These defects, in turn, can cause secondary flaws, such as the presence of voids or alterations in thickness [9]. These different types of defects appear particularly critical if the components are subject to high internal pressures, as happens in the case of tanks or pipes for pressurized fluids; therefore, the issue of identifying defects for product quality control, even during manufacturing, or for process optimization purposes, appears particularly important for safety reasons.

For these reasons, in this paper, some solutions for analyzing the fiber orientation in a carbon fiber cylinder, obtained through a filament winding process, are explored, highlighting, in particular, the metrological aspects involved in this type of investigation.

There are presently a lot of non-destructive techniques for detecting damage on composite structures like thermography, acoustic, ultrasonic, eddy current techniques, computed tomography imaging, and X-rays [5,10]. Computed tomography (CT) is one especially advanced non-destructive technique that offers geometrically precise subsurface imaging of a component, useful for visualizing and evaluating the fiber characteristics on CFRP components [11,12,13]. One drawback of this technique is that the complete component scan is frequently unachievable due to the physical size of typical aerospace components [11]. Furthermore, even in the most modern versions, CT scanners continue to be expensive and cumbersome.

In general, all the cited technologies allow for the study of the completed part or after failure, but they cannot provide guidance or support during the assembling process [14].

Instead, vison-based methods are useful instruments for the online inspection of pieces [15,16]. The drawback of these techniques for CFRP inspection is that they are challenging to use in the case of non-optimal illumination conditions and, as happens in many cases, glossy surfaces [1]. Many approaches based on edge detection and conventional image processing are proposed, but these need high-resolution and quality pictures, which are not always easily achievable, taking into account operational and surface characteristics [8,17].

It has recently been shown that carbon fibers alter the direction of the reflected light’s electrical field, so that in each point of the surface of a CFRP piece, the angle of polarization of reflected light matches the fiber orientation; this phenomenon allows to use a particular branch of the computer vision, the polarization technique, to analyze the characteristics of the CFRP weave [1,8,18].

Polarization vision has become a well-established field overall, and it has long been employed in photography to prevent gloss and mirror effects and to improve visibility [19], and, in more recent applications, it has also been used in high-dynamic-range (HDR) imaging [20], in the orientation and navigation fields [21], for characterizing birefringent fluids [22], and for studying acoustical phenomena [23]. Its application to the particular subject of composite component inspection is still relatively new [1,24].

Modern sensors exist, which collect polarized light in four planes (at 0°, 45°, 90°, and 135°) using microscopic polarizing filters on each pixel. Based on the intensities obtained for each pixel, related to the polarized light in the four directions, it is possible to compute, through Stokes parameter analysis [10], the angle of linear polarization (AoLP), which aligns with the fiber angles on the surface under investigation. If the incident light is assumed to be unpolarized, then any induced polarization in the observed reflected light is solely caused by the reflected light’s electric field aligning with the fibers. This property is still present when resin is applied on the carbon fiber component.

This method, compared to non-vision and other vision-based methods, on average presents the main advantages and drawbacks shown in Table 1 [1].

Given the interest in online inspection methods of fiber orientation for composite material components, the authors intend to delve into the metrological aspects related to the use of the polarized camera for the measurement of the winding angle.

The approach followed in this article for the fiber angle measurements is similar to that described by [1], which involves analyzing the AoLP distributions and evaluates the distance between peaks to determine the angle between tows.

In [1], however, the variability analysis is based on the evaluation of the standard deviation of the identified Gaussian distributions, which is also affected by dust or loose fibers on the surface. This assessment is believed to be an overestimation of the uncertainty, considering that the angle measurements are obtained by averaging a large number of values, equal to the number of pixels corresponding to that specific angle. In addition, the absence of a reference for the fiber angles makes it difficult to evaluate errors, which are assessed relative to nominal values. Furthermore, the paper identifies the aspects of greatest influence on the measurement: lighting mode, camera settings, and surface curvature. However, a structured approach to evaluating these factors’ effects and interactions is missing.

In this paper, a design of experiment (DOE) was carried out to estimate the combined effect of exposure time, lens aperture, and distance of the lighting system, with the aim of identifying the optimal system settings and estimating their impact on the variability of measurement. Furthermore, an uncertainty assessment was carried out combining the main uncertainty contributions.

This metrological analysis is deemed innovative by the authors with reference to the inspection by polarization, compared to the state of the art in this field.

In addition, in the present work, a study of the trend of the winding angle along the surface of the cylinder under analysis was carried out, in order to obtain useful information for the optimization of the production process.

## 2. Materials and Methods

### 2.1. Vision System

A FLIR Blackfly S BFS-PGE-51S5P camera (Flir, Wilsonville, Oregon) is used to capture the component surface. This monochrome polarization camera is based on the Sony IMX 250 CMOS sensor (Sony, Tokyo, Japan) and has a native resolution of 5.0 Mpixels (2448 (H) × 2048 (W) pixels). The pixel size is 3.45 µm and incorporates a four-direction polarizer formed on the photodiode of the chip. With this special construction, it is possible to simultaneously capture the polarization information in four different directions. Microscopic filters are present on each pixel with polarization directions as described in Figure 1; from top left to bottom right, they are 90°, 45°, 135°, and 0°.

To trade-off between the working distance and minimizing the perspective distortion, 25 mm low distortion lens are used in this paper.

Two soft boxes equipped with 80 W white bulbs are employed as light sources. With this setup, it is possible to illuminate the component under analysis with a diffused and non-polarized light. This kind of illumination system is chosen for this application due to the dimensions of the component under analysis. To reduce the ambient light influence, the component is placed inside a white box and away from ceiling lights.

The working distance of the camera is set to 190 mm. This provides a good trade-off between the extension of the acquired surface and the influence of the curved shape of the component of interest; this aspect will be discussed in depth in Section 2.4.

Images are acquired in RAW file format to preserve all the information from the camera sensor using Spinnaked SpinView (version 2.2.0.48) acquisition software. Figure 2a shows a scheme of the acquisition setup where the softboxes are placed at Sd1 and Sd2 distances, respectively, from the component surface, while Figure 2b depicts the developed vision system with camera, illuminators, and the component under analysis.

### 2.2. Component under Analysis and Inspection Strategy

This work focuses on the study of a non-planar component, in particular, a composite cylinder that represents a part of a pressure vessel. This part is made by means of a filament winding process using carbon fiber and epoxy resin towpregs.

All the materials used in the manufacturing process are commercially available. For the carbon fiber, Toray T700S-12K-50C was used, whereas the resin system was a Huntsman Araldite Solution, composed of Araldite LY 3508 as epoxy resin, Aradur 1571 as hardener, and Accelerator 1573 as accelerator [9].

The obtained tows are wound in layers on a stainless steel rotating mandrel. The winding pattern consists of an initial hoop layer where the tows are wound with a winding angle close to 90° to promote adhesion of the subsequent helicoidal layers.

The quantity of interest to be measured is the winding angle, which can be obtained starting from the angle between the tows (2β), as the difference between 90° and the angle β, as explained in Figure 3.

The nominal winding angle for these last layers is 60°.

The cylinder under analysis has an internal diameter of 200 mm and a total length of 300 mm. To study the component’s surface, it is inspected along two longitudinal directions and three circumferential lines. The longitudinal directions, L1 and L2, are spaced about 70° from each other. For the circumferential lines, one, C2, is positioned on the central section of the component, while the others, C1 and C3, are placed 65 mm apart on the left and on the right of the previous one. The component and its subdivision are reported in Figure 4.

Following the considered division of the surface, along the longitudinal directions, 4 areas are examined, while on the circumferential lines, 22 areas are analyzed. The test plan provides three repeated acquisitions for each considered area of the cylinder, repositioning the component each time.

### 2.3. DOE for the Optimal Setting of the Vision System

The setting condition under which to carry out the tests are established in advance using a DOE which minimizes the variability of the results.

Optimizing and characterizing a vision system for the application of interest is not trivial since several parameters influence the result. For this reason, a DOE approach is employed to systematically study the effects and the relationship between different factors. In this case, three parameters are investigated: the distance between the soft boxes and the component under analysis, exposure time, and lens aperture. The lighting system distance defines how much light illuminates the component surface. The exposure time is the amount of time the camera sensor records the light. Using low times, the pixels are very dark, resulting in images with poor signal to noise ratio (SNR). On the other hand, longer exposure could lead to overexposure and saturations on the image. Lens aperture directly affects the amount of light that enters the camera sensor, thus influencing the overall brightness of the photo.

A 2^3^ experiment is designed, where two levels are considered for each factor, as reported in Table 2. With this experiment design, eight different factor combinations are possible resulting in as many settings. Each of them is independently replicated three times considering the same area on the cylinder surface. Furthermore, the experiment runs are performed in random order.

The experiment results are compared considering the variability of the measured fibers angle that is chosen as desired output (the so called “response”) of the DOE. In this case, the aim of the DOE is to identify the configuration that minimizes the response. Once the optimal settings for the vision system are identified, the cylinder surface is analyzed.

It should be noted that in this metrological analysis the following aspects of influence were kept constant and therefore their effects were not analyzed:the type of illumination, which was chosen as diffuse and kept constant, once the optimal distance was identified;the type of lenses, in particular the focal length, was also kept constant in this analysis (25 mm);the type of material and the piece geometry.

### 2.4. Data Processing Procedure

The acquired images are processed using MATLAB^®^, release 2023b.

The data processing procedure for the measurement of the angle between tows consisted of the following steps:
To exploit the extra information a polarization camera provides, the first pre-processing step was to “demoisaic” the captured images. From the RAW file format following the scheme provided in Figure 1, the pixel intensity for each direction is extracted from the original acquisition. As a result of this step, there were four images, one for each polarization direction. The resulting images have a resolution of 1224 (H) × 1024 (W) pixels, one fourth of the original size. The reduction in the final image resolution is a negative aspect of this kind of polarization camera.From the pixel intensity Iθ for each direction θ (0°, 45°, 90°, and 135°), the Stokes parameter are calculated, as classically indicated by the scientific literature in the field [1,14,25]:(1)$$S_{0}=\frac{I_{0}+I_{45}+I_{90}+I_{135}}{2}$$
(2)$$S_{1}=I_{0}-I_{90}$$
(3)$$S_{2}=I_{45}-I_{135}$$From the Stokes parameters, it is possible to calculate the degree of linear polarization (DoLP), which represents the ratio of the intensity of the polarized to the intensity of the unpolarized part of the light, ranging from 0 (for unpolarized light) to 1 (for totally polarized light). In particular, DoLP is calculated using the following equation:
(4)$$DoLP=\frac{\sqrt{S_{1}^{2}+S_{2}^{2}}}{S_{0}}$$As pointed out in [1], DoLP can be used to ensure that the light is sufficiently polarized to maintain acceptable SNR.In the case study, to reduce the effect of the complex shape of the component, the DoLP is used to identify in the image the region of interest (RoI) in which to carry out the angle measurement. Taking into account indications from the scientific literature [1], which identify DoLP values of about 0.4 as acceptable, the RoI is chosen by identifying the area in which the number of points with DoLP greater than or equal to 0.4 is prevalent.From the Stokes parameters, the angle of linear polarization (AoLP) is calculated in the RoI identified at the previous step, using the following equation:
(5)$$AoLP=\frac{1}{2}\mathrm{atan}\frac{S_{2}}{S_{1}}$$In this case, the AoLP function is calculated using the inverse tangent function that returns values in the range [−90; 90] (angle measured from the vertical direction, clockwise). The AoLP aligns with the fiber angles on the surface under investigation.The relative frequency distribution of the AoLP in the RoI is studied. It should be distributed according to a bimodal distribution, because there are two prevailing directions of the fibers constituting the winding of the composite material. For verification purposes, in this work, the bimodality characteristic is studied using the bimodality coefficient (BC) [26]:
(6)$$BC=\frac{S^{2}+1}{K+3\frac{{\left(n-1\right)}^{2}}{\left(n-2\right)(n-3)}}$$where S is the skewness, K is the kurtosis, and n is the sample size. If the value of this index is more than the critical value of 5/9 ≃ 0.56, it suggests bimodality of the distribution [26].Other metrics are considered to further evaluate the characteristics of the bimodal distribution, in particular, the bimodality amplitude (BA) and the bimodality ratio (BR) [27]. BA is defined as the ratio of the smaller peak amplitude to the antimode amplitude of the fitted probability density function (PDF). It is always less than or equal to 1, and larger values indicate more distinct PDF peaks. BR is the ratio of the right and the left amplitude peaks of the PDF, indicating which of the two dominates.Then, in the case of the cylinder, for each image acquired, bimodality confirmed, the AoLP data are plotted on a histogram and are fitted using the sum of two Gaussian distributions.Finally, the mean angle between tows (2β) is evaluated by identifying the angles corresponding to the peaks of the two fitted Gaussian distributions and calculating their difference (the distance between them).It is important to emphasize that the angles corresponding to the peak of each Gaussian are average values of the angles obtained in all the pixels of the image corresponding to that direction. Therefore, the winding angle that is obtained is an average value over the RoI analyzed; this average value represents the measurand in this paper.

### 2.5. Uncertainty Assessment Methodology

The uncertainty assessment is carried out considering the following components: Repeatability of the measurements on the same area: It is evaluated by conducting three repeated measurements for each examined area, on three different images. In each acquisition, the component is repositioned. Then, the maximum difference D between results is calculated and the uncertainty contribution is determined, considering a rectangular distribution as $D/\left(2\sqrt{3}\right)$Variability of the measurement on the surface analyzed: Standard deviation calculated on the bases of the measurements made along the circumferential lines of the cylinder, since in these directions, during the winding process, the pulling forces of the deposition head can be considered constant. For this reason, theoretically, all the tows should be wound at the same angle on the rotating mandrel. This contribution comprises the variability of the measurements along the circumferential lines.Contribution of the setting parameters: The influence of the parameters is evaluated as the standard deviation of the mean values of the measured tow angles obtained in the different acquisition setups considered in the DOE. The largest difference D’ in winding angles is identified, and the contribution to the uncertainty is assessed, considering a rectangular distribution as $D\prime /\left(2\sqrt{3}\right)$

The shape of the object under analysis should be considered as another source of uncertainty since the reflected light polarization is also influenced by the curvature of the surface. For this reason, the camera axis is positioned in radial direction with respect to the cylinder, and the measurement is performed in a central band of the image (width ~20 mm), which was identified taking into account the highest DoLP values, as described in Section 2.4. In this way, the curvature effect is expected to be negligible with respect to the other contributions, also considering the limited extension of the inspected area compared to the diameter of the component (200 mm), so each area examined can be considered nearly flat.

## 3. Results

The results of the described methodology, applied to the cylinder under test, are organized as follows:Results of the DoE and setup definition;Setting of the RoI dimension, through the DoLP analysis of one of the acquired images;Winding angle measurement on the same image considered in the previous point, through the AoLP frequency distribution analysis;Uncertainty assessment;Analysis of the winding angle along the surface.

### 3.1. DOE and Acquisition Setup Definition

As stated in Section 2.3, the desired output (“response”) for the DOE is the variability of the fiber angle; in particular, the standard deviation of the fitted Gaussian with the higher peak was taken as a reference parameter.

The DOE approach highlights the interactions between factors represented in Figure 5, changing their level from the lower value (indicated by “−1”) to the higher value (indicated by “1”) as reported in Table 2.

The factors that show a strong interaction are exposure time and lens aperture. On the contrary, the analysis of the distance and exposure time combination does not indicate possible interactions since the two responses lines are parallel [28].

An analysis of variance (ANOVA) confirms the suggestions given by the two-factor interaction plots; that is, the parameters with the most impact are exposure time and lens aperture and their interaction.

The standard deviation taken as the response of the DOE is minimized in correspondence to the minimum distance, the maximum exposure time, and the minimum lens aperture. This result is understandable, considering that a lower distance of softboxes and a greater exposure time increase the brightness of the picture, and a wider aperture raises the variability, probably due to the reduced depth of field. The reduced depth of field, in fact, could cause different levels of image defocusing due to the non-planarity of the surface and variability in positioning at the working distance.

In the light of these results, the vision system is optimized for the application of interest by choosing the maximum exposure time and the minimum lens apertures, which are 150 ms and f/8, respectively, while the two illuminators are positioned closer to the surface, at a distance of 400 mm.

### 3.2. Setting of the RoI

After the Stokes parameters calculation, for the setting of the RoI dimension, the first analysis step is to evaluate the DoLP in the acquired image. This indicates the amount of captured light that is polarized. As pointed out in [1], the DoLP can be used to ensure that the light is sufficiently polarized to maintain an acceptable SNR. One of the images acquired on the cylinder, on the central circumferential line C2, was used for this purpose. Figure 6 reports the DoLP of the selected region of the cylindrical surface.

As shown in Figure 6, the DoLP is not uniform on the whole component surface, due to its curvature, and the highest values (greater than or equal to 0.4) in particular are in the central band, so the image could be ideally subdivided into three vertical zones. Then, the analysis of the winding angles was carried out on the central band, 20 mm wide, where a number of points with a DoLP greater than or equal to 0.4 are prevalent.

It is interesting to note that the component curvature also influences AoLP values, as can be seen in Figure 7, which shows a color map of the AoLP of the same area.

Figure 7b is a close-up of the central part of the AoLP image, where it is possible to see some pixels with different colors that generate noise and tow angle variability, due to resin irregularities and particles or loose fibers on the surface that affect the measurement.

### 3.3. Winding Angle Measurement

On the bases of the Stokes parameters, considering the image’s central region, with dimensions of 408 × 1024 pixels, the AoLP value is calculated in each pixel. The frequency distribution of the AoLP in the RoI identified at the previous step is represented in Figure 8.

The BC is calculated for the AoLP values of the central RoI to ensure that the distribution is bimodal. It is equal to 0.59, meaning a bimodality distribution. Furthermore, the metric BA is equal to 0.93, signifying distinct PDF peaks; BR is 0.66, indicating that the left peak is greater than the right one, as evident in Figure 8, due to the prevalence of pixels characterized by a polarization angle of approximately −31° (angle measured from the vertical direction).

In Figure 9, the histogram of the AoLP is fitted by the sum of two Gaussian distributions.

For the first Gaussian, the mean value is equal to −30.8° with a standard deviation of 6.1°. For the second Gaussian, the mean value is 28.9° and the standard deviation is 7.3°. From the mean values of the Gaussians, the angle between tows (2β) is calculated by difference. As described in Section 2.2, to evaluate the winding angle, half of the tows angle is subtracted from 90° and the result is equal to 60.2°.

### 3.4. Uncertainty Assessment

The uncertainty contributions considered, as explained in Section 2.5, are as follows:Repeatability of the measurements in the same area;Variability of the measurement on the surface analyzed;Contribution of the setting parameters.

The results of the uncertainty assessment, carried out as described in Section 2.5, are summarized in Table 3.

The variability of the fitting algorithm is considered negligible since the committed error is very low using synthetic data.

Summing the square of all components under the square root, the resulting standard uncertainty equals 0.5°. Considering a nominal winding angle of 60°, the percentage uncertainty is in the order of 0.8%.

It is important to underline that this overall uncertainty tends to overestimate the uncertainty of the measurement method, because it also takes into account the variability of the measurements, due to the irregularities of the process. More precisely, it represents the uncertainty with which the winding angle is measured on the specific examined component, and which must be taken into account to detect any variations along the longitudinal direction that may be considered significant.

It is interesting to note that the uncertainty obtained is lower than the one obtained in a previous work by the authors [29] on the same cylinder, but using a classical analysis method, based on a traditional color camera and an edge detection algorithm. In that case, in fact, the assessed standard deviation was of the order of 1.7°. Furthermore, the algorithm for determining the angle between tows required the manual definition of RoIs and was not easily automatable unlike the method presented in this work.

### 3.5. Winding Angle along the Surface

Figure 10 represents the winding angle along the three circumferential lines of the cylinder (C1, C2, and C3). The error bars represent the overall uncertainty, as standard deviation, of the winding angle reported in Table 3, for each measuring area.

The winding angle along the longitudinal directions is reported in Figure 11.

Figure 10 shows that along the circumferential directions, the winding angle remains substantially constant, apart from irregularities attributable to variability in the process. Furthermore, as shown in both Figure 10 and Figure 11, the winding angle along the longitudinal direction tends to increase in the cylinders’ external parts, reaching a maximum value of 70°, while in the central areas of the cylinder, the winding angle is near to the nominal angle of 60°. This evidence suggests problems of the winding process in the outer parts of the component, which could lead to defects inside the structure such as tows folding.

## 4. Discussion

In the light of the obtained results, the most significant observations can be summarized as follows: –The uncertainty evaluated (u), equal to 0.5° as standard deviation, appears satisfactory and adequate for the application, since it allows to distinguish variations along the surface of the cylinder, which can be attributed to problems of the production process. In fact, along the longitudinal direction, significant variations in the winding angle are highlighted. In particular, at the ends of the cylinder, the angle increases by 5–10° compared to the central values (Figure 11); these increments are much higher than the uncertainty, even considering the expanded uncertainty at a 95% confidence level (U = 2 × 0.5° = 1°). The differences in the winding angle along the cylinder can be interpreted as a problem in the relative movement of the deposition head and the rotating mandrel; this information could be useful for production process optimization.–The uncertainty of the described method is lower than that obtained using a classical analysis method, based on a traditional color camera and an edge detection algorithm in a previous work by the authors [29] on the same cylinder. In that case, in fact, the assessed standard deviation was of the order of 1.7°.–With respect to other methods of image analysis like that mentioned in the previous point, the polarization-based method can be easily automated and implemented online for quality control during manufacturing. In fact, the algorithm does not require any manual intervention, but obtains the average winding angle within the identified RoI automatically.

## 5. Conclusions

This paper presents a procedure to study the winding angle of a cylinder made of carbon fiber and epoxy towpregs wound in a helical pattern. The winding angle was assessed using a vision system based on a polarization camera. The image post-processing exploited the extra information provided by this polarized sensor, and the data are analyzed using a statistical approach.

The measurement method was studied from a metrological point of view, and the uncertainty analysis showed that using a polarization vision system on the analyzed tow-wound cylinder provides an uncertainty in the winding angle measurement of 0.5°.

The analysis of the component’s surface revealed that along the circumference of the cylinder, the winding angle is stable and close to the nominal value of 60°, whereas there are significant variations in the ending parts of the component, taking into account the uncertainty value. In particular, on the ends, it reaches a maximum value of 10° more than the designed value. This difference of the winding angle can be interpreted as a problem in the relative movement of the deposition head and the rotating mandrel.

In conclusion, the vision technique based on the light polarization has proven appropriate for characterizing the manufactured component and giving helpful information about the manufacturing process for the purpose of optimization. These findings also motivate further research into the use of polarization cameras for surface inspection and winding angle measurement of woven carbon fiber components.

## Figures and Tables

**Figure 1 sensors-24-05685-f001:**
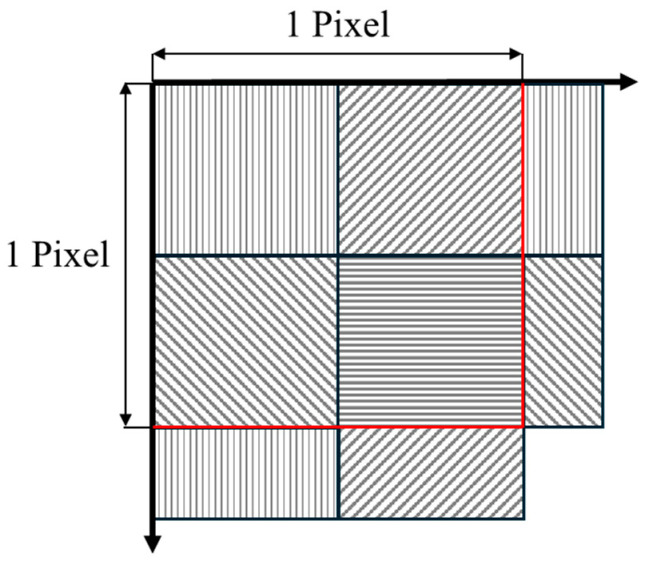
Scheme of the polarization filter directions on the sensor.

**Figure 2 sensors-24-05685-f002:**
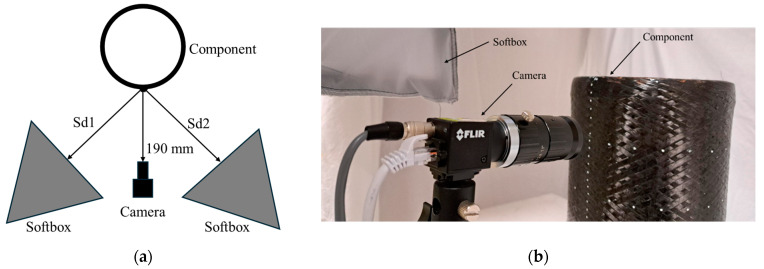
Acquisition setup: (**a**) Scheme (top view); (**b**) Picture of the developed vision system.

**Figure 3 sensors-24-05685-f003:**
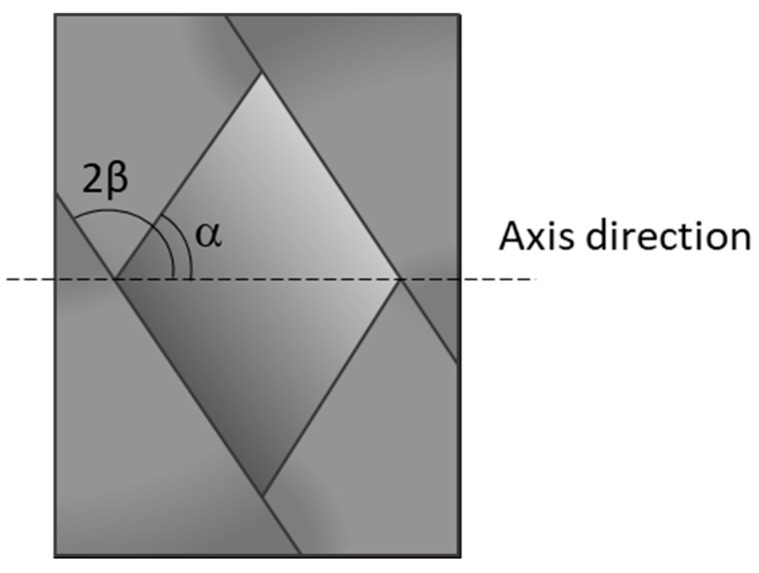
Angles in a helicoidal winding pattern.

**Figure 4 sensors-24-05685-f004:**
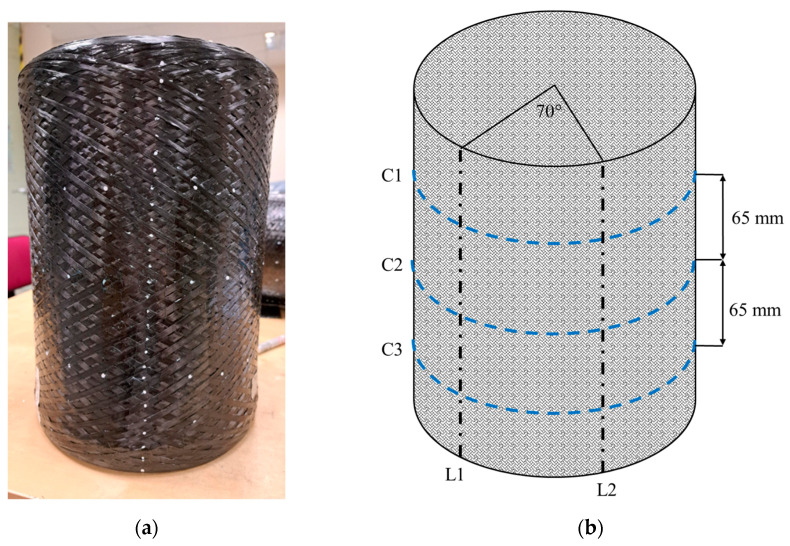
Cylinder used in the present work: (**a**) Picture; (**b**) Scheme of the surface subdivision.

**Figure 5 sensors-24-05685-f005:**
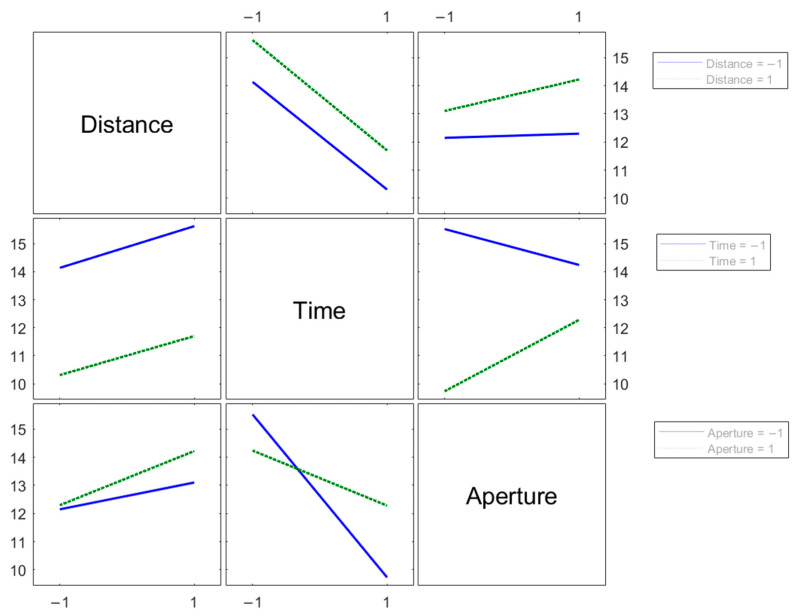
Plot of the interactions between two factors.

**Figure 6 sensors-24-05685-f006:**
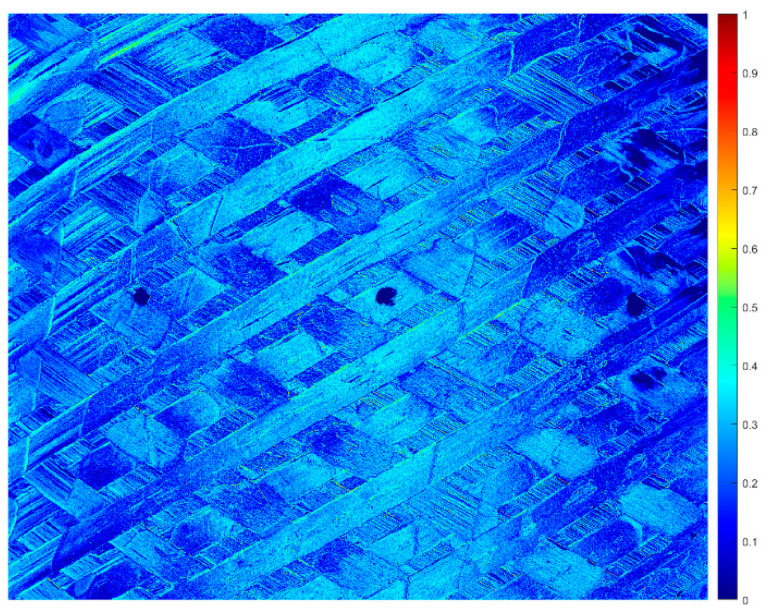
DoLP on the acquired area of the surface.

**Figure 7 sensors-24-05685-f007:**
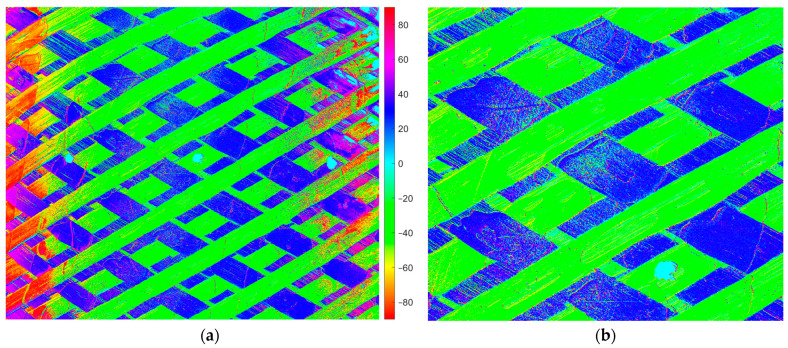
Example of an AoLP image: (**a**) Color map to highlight the different angles; (**b**) Close-up of the AoLP image shows the refraction near the resin.

**Figure 8 sensors-24-05685-f008:**
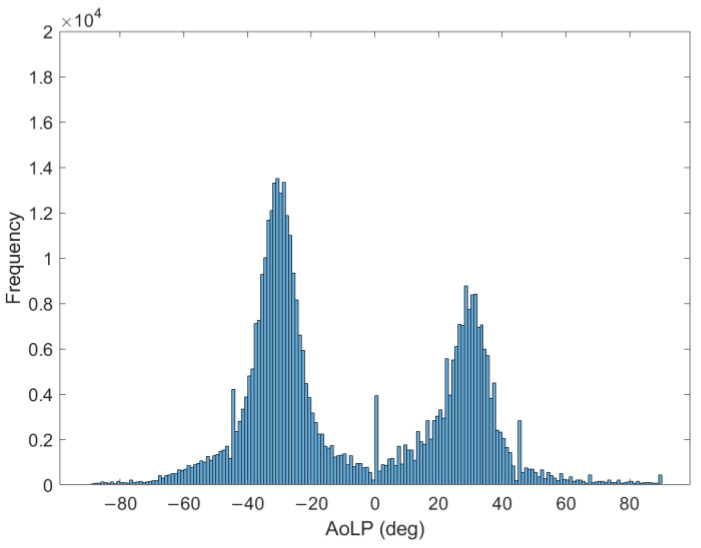
Histogram of the AoLP values.

**Figure 9 sensors-24-05685-f009:**
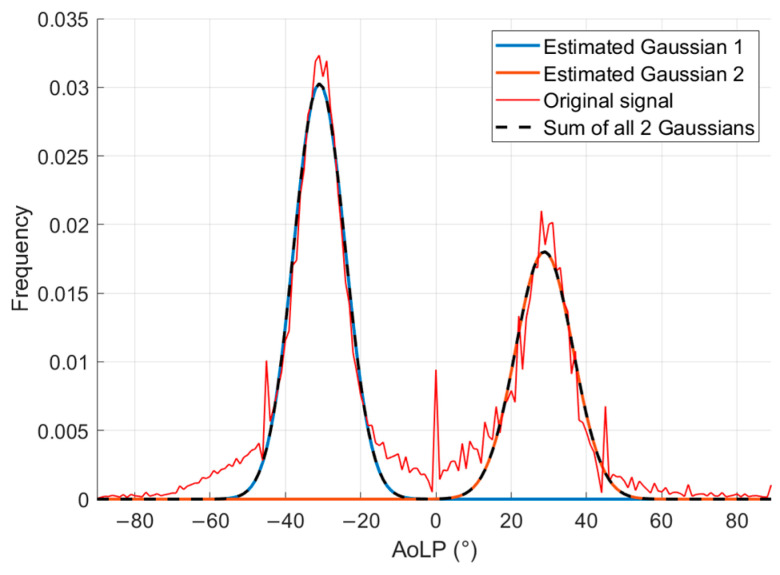
Two Gaussians used to fit the distribution of the AoLP.

**Figure 10 sensors-24-05685-f010:**
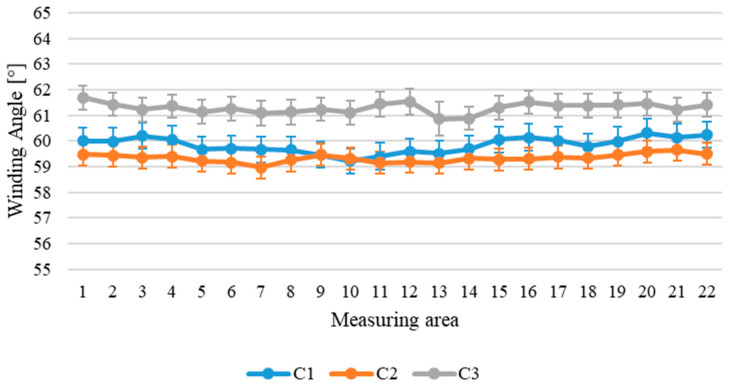
Winding angle along the circumferential sections of the cylinder.

**Figure 11 sensors-24-05685-f011:**
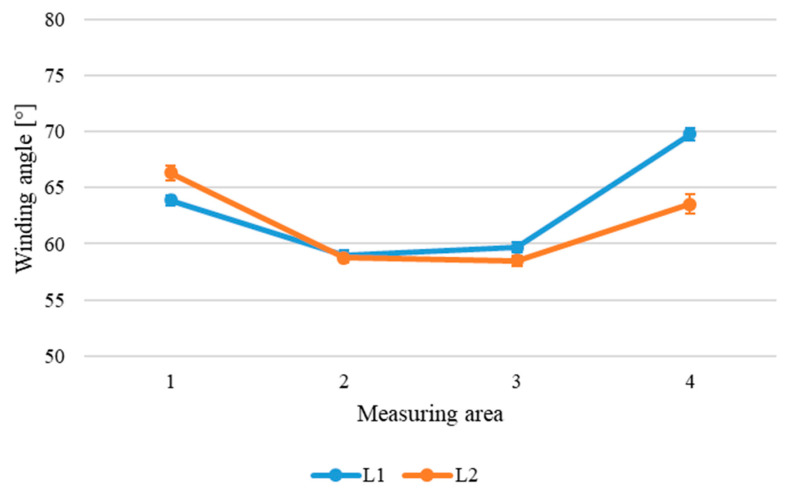
Winding angle along the longitudinal directions of the cylinder.

**Table 1 sensors-24-05685-t001:** Advantages and drawbacks of polarization vision compared to other methods for composite materials inspection.

Compared to Non-Vision Methods	
Advantages	Drawbacks
cost-effectiveness ease of installation on a manufacturing line greater resolution lower consumption faster data acquisition	inspection limited at the uppermost visible layer unsuitability for glass fibers
Compared to other Vision-Based Methods	
Advantages	Drawbacks
more robust to inadequate lighting efficacious even on unidirectional surfaces and different structures or curing states of the surface ease of automation of the processing algorithm	lower resolution slightly more costly camera than traditional ones

**Table 2 sensors-24-05685-t002:** Experimental factors for the characterization of the vision system and their levels.

Factor	Level	
	Low	High
Softbox distance [mm]	400	600
Exposure time [ms]	15	150
Lens aperture [mm]	f/8	f/4

**Table 3 sensors-24-05685-t003:** Uncertainty components for the winding angle measurement.

Factor	Distribution	Uncertainty Contribution [°]
Repeatability	Rectangular	0.2
Measurand	Gaussian	0.3
Parameter setup	Rectangular	0.3
Overall standard uncertainty (u)		$\sqrt{{\left(0.2\right)}^{2}+{\left(0.3\right)}^{2}+{\left(0.3\right)}^{2}}=0.5$

## Data Availability

The original contributions presented in the study are included in the article, and further inquiries can be directed to the corresponding authors.

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
