# Peer review of "Using Light Polarization to Identify Fiber Orientation in Carbon Fiber Components: Metrological Analysis"

_sensors, 2024, doi:10.3390/s24175685_

Round 1

Reviewer 1 Report

Comments and Suggestions for Authors

The authors of this paper presented a method for measuring tow angles in carbon fiber components by using a polarized camera. The method leverages the unique property of carbon fibers that alters the direction of reflected light, allowing the alignment of the angle of polarization with the fiber orientation. By performing a statistical analysis of the angle of linear polarization (AoLP) across the component's surface, they were able to determine the average winding angle. Different from previous pioneer work, authors explored the metrological aspects of this approach, focusing on the measurement uncertainty and optimization of the process parameters, such as exposure time, lens aperture, and lighting distance, with the goal of reducing variability in the measurements. The results demonstrated the method's effectiveness in evaluating the distribution of angles on the surface of carbon fiber reinforced polymers. Overall, the manuscript clearly presented the method, design of experiments and results discussion. Some minor issues I think can be addressed to further improve the quality of this publication. I suggest the manuscript to be accepted by sensors after a minor revision.

Minor:

1.      L96-L112, I understand authors want to introduce the pros and cons of the polarization based method. However, the logic is kind of confusing here. E.g., L06-L102, L107-110 are about pros and L103-L106, L111-L112 are about cons. It would be better if author can group them just into two sections, advantages and drawbacks. That would be much more clear to authors.

2.      L325, the shallower depth of field will make sure that only the in-focus target contribute to the pictures. I’m wondering why that will cause the larger variation. Authors may want to explain a little bit more about this observation.

3.       L310-311, the language here is obscure to me. Several shorter sentences will make it much more readable and clearer.

4.      L346-347, how is this 20mm wide band is chosen? Is it an arbitrary number? Since the Figure. 6 already provides a quantitative result on the DoLP, shall we consider to provide a more solid/logical process on determining the sampling area.

5.      L349-350, the color bar of Figure 7 is missed.

Reviewer 2 Report

Comments and Suggestions for Authors

This manuscript presents a procedure to study the winding angle of a cylinder made of carbon fiber and epoxy towpregs wound in a helical pattern. However, further improvement is needed before publication. There are still the following problems:

1. Please refine the steps of the experiment and add appropriate experimental details.

2. Please add any unique aspects of applied metrology that are not highlighted in the study.

3. Lack of comparison of body measures with other methods in the study.

4. The discussion section is weak in the discussion and analysis chapters and it is recommended that the results of the experiments be fully discussed.

5. Measurement methods are emphasized throughout, but they are not reflected in the materials and methods.

Reviewer 3 Report

Comments and Suggestions for Authors

The manuscript is a work of interest both for the potentiality of new materials based on CRPF and for the need for methods and technologies for NDT in these materials.

The manuscript is well constructed, the experiments are well told, and the results achieved are interesting.

I recommend publishing it without changes since I have not found any typos.

Author Response

Comments 1: The manuscript is a work of interest both for the potentiality of new materials based on CRPF and for the need for methods and technologies for NDT in these materials. The manuscript is well constructed, the experiments are well told, and the results achieved are interesting. I recommend publishing it without changes since I have not found any typos.

Response 1: The authors are very grateful to the referee for the valuable review of the work and for the appreciation.